# Rasagiline Inhibits Human Melanoma Cell Viability and Interacts Synergistically with Mitoxantrone and Antagonistically with Cisplatin—In Vitro Isobolographic Studies

**DOI:** 10.3390/cancers17152563

**Published:** 2025-08-03

**Authors:** Danuta Krasowska, Paula Wróblewska-Łuczka, Michał Chojnacki, Katarzyna Załuska-Ogryzek, Jacek Kurzepa, Jarogniew J. Łuszczki

**Affiliations:** 1Department of Medical Chemistry, Medical University of Lublin, 20-093 Lublin, Poland; dana.krasowska@gmail.com (D.K.); jacek.kurzepa@umlub.pl (J.K.); 2Department of Occupational Medicine, Medical University of Lublin, 20-090 Lublin, Poland; katarzyna.zaluska-ogryzek@umlub.pl (K.Z.-O.); jarogniew.luszczki@umlub.pl (J.J.Ł.); 3Department of Medical Biology, Institute of Rural Health, 20-090 Lublin, Poland; chojnacki.michal@imw.lublin.pl

**Keywords:** human malignant melanoma, rasagiline, cisplatin, mitoxantrone, isobolographic analysis

## Abstract

The increased incidence of malignant melanoma is observed in patients with Parkinson’s disease. Rasagiline, but not carbidopa produced the clear-cut anti-proliferative effects on various melanoma cell lines. Rasagiline combined with mitoxantrone exerted the most desirable synergistic interactions in relation to the anti-proliferative effects in four malignant melanoma cell lines, as assessed isobolographically. In contrast, rasagiline should not be combined with cisplatin during the treatment of malignant melanoma due to the antagonistic interactions in the MTT assay.

## 1. Introduction

Parkinson’s disease (PD) is the second most common neurodegenerative disease, affecting more than 10 million people around the world [1,2]. Epidemiological evidence indicates that patients with PD are at increased risk of developing melanoma [3,4,5]. Interestingly, the risk of PD is greater in people who have a family history of melanoma among close relatives [6]. A recent meta-analysis estimated that the increased risk of melanoma is approximately twice as high among people with PD [7]. Another meta-analysis based on 63 studies involving approximately 17 million patients also confirmed that PD was significantly associated with a 75% increased risk of melanoma. However, there was no significant association between PD and non-melanoma skin cancer [8]. The incidence of melanoma in patients with PD ranges from 1.1 to 1.4% [9,10].

The interdependence of PD and melanoma is likely because mutations or other changes in many genes and proteins are common to PD and melanoma, including factors contributing to melanin biosynthesis, cellular detoxification, the oxidative stress response, and cellular transport pathways that could combine these two diseases [11,12,13,14]. An example is the relationship between the regulation of pigmentation by the MC1R (Melanocortin 1 receptor) gene, which occurs in both melanocytes and dopaminergic neurons [12,13]. Other genetic mutations, such as CYP2D6 (Cytochrome P450 2D6) polymorphisms and VDR (Vitamin D Receptor) polymorphisms, may also be associated with both diseases [14].

Evidence suggests that some of the drugs used to treat PD contribute to an increased risk of melanoma. One of such drugs is levodopa (L-DOPA), which is a precursor of both dopamine and melanin [15,16]. Some reports show an increased risk of newly developing PD after a previous diagnosis of melanoma, suggesting that medications are not involved [5,11]. Moreover, the association between L-DOPA and melanoma, apart from the common biochemical pathways of dopamine and melanin synthesis, is based on several case reports, which do not demonstrate a cause-and-effect relationship [9,17]. Moreover, levodopa has not been shown to have a carcinogenic effect [18]. Another hypothesis of the interdependence of both diseases is that people with PD are more sensitive to sunlight, and long-term exposure to ultraviolet (UV) radiation often leads to the development of melanoma. L-DOPA or other drugs may be photocarcinogenic [9,19,20].

During clinical trials with rasagiline, a renewed interest in the connection between PD and melanoma occurred when single cases of melanoma were reported by clinicians [21]. Rasagiline binds to melanin, and its mechanism of action involves irreversible inhibition of monoamine oxidase subtype B (MAO-B) [22,23], increasing extracellular levels of dopamine, which is important in the treatment of PD [22,23,24,25]. Safety comparisons among MAO inhibitors against PD revealed some cases of newly diagnosed malignancies in patients taking rasagiline. In the TEMPO study, 3 out of 132 patients in the group receiving a dose of 2 mg of rasagiline were diagnosed with malignancies, including melanoma, prostate cancer, and squamous cell carcinoma. Moreover, in the second phase of the study, 5 out of 371 patients had newly diagnosed cancers (colon cancer, two cases of squamous cell carcinoma of the skin) and the risk assessment for using rasagiline resulted in the recommendation of periodic dermatological examinations of patients taking this drug [25,26,27,28,29].

All these conflicting reports regarding drugs used in PD and their possible impact on the development of melanoma influenced the experiments presented herein. The aim of this study was to evaluate the effect of carbidopa (DOPA decarboxylase inhibitor) and rasagiline (an inhibitor of MAO-B) on the viability and proliferation of four human melanoma cell lines (FM55P, A375, FM55M2 and SK-MEL28). To compare the results, the experiments were also conducted on a normal human keratinocyte cell line. Furthermore, the effect of rasagiline on cell cycle arrest of human melanoma cells was verified so as to confirm its crucial role in human melanoma. It was assumed that if any of the tested compounds affect the viability of melanoma cell lines, and the obtained results allow for the calculation of the IC_50_ value, the compound will be subjected to combinations with chemotherapeutics. Therefore, the pharmacological profiles of interactions of rasagiline with two classic chemotherapeutics, cisplatin (CDDP) and mitoxantrone (MTX), were assessed isobolographically with respect to the cell viability in FM55P, A375, FM55M2, and SK-MEL28 cell lines. It is of great clinical importance to verify whether the drugs used in PD treatment can affect melanoma. Furthermore, for these patients, multi-drug therapies are being sought as they offer significantly better outcomes than monotherapy.

## 2. Materials and Methods

### 2.1. Cell Culture

Primary (FM55P, A375) and metastatic (FM55M2, SK-MEL28) malignant melanoma cell lines were used in the experiment. From ECACC, we purchased two cell lines: FM55P and FM55M2 (European Collection of Cell Cultures, Public Health England, Porton Down, UK). Two another cell lines (A375 and SK-MEL28) were purchased from ATCC (American Type Culture Collection, Manassas, VA, USA). We used the following media in cell culture: RPMI- 1640 Medium, DMEM- high glucose, and EMEM (all from: Sigma-Aldrich, St. Louis, MO, USA). More information about cells and culture conditions was described elsewhere [29,30,31].

After 24 h of incubation of cells, the medium was replaced with fresh medium, to which increasing concentrations of rasagiline, carbidopa, MTX, and CDDP were added. Next, MTT, LDH, and BrdU assays were performed as described below. After obtaining the preliminary results, an isobolographic analysis was conducted.

### 2.2. Drugs

Carbidopa (stock 20 mM) and mitoxantrone (MTX) (stock 10 mM) were dissolved in DMSO, whereas rasagiline mesylate (stock 50 mg/mL) and cisplatin (CDDP) (stock 1 mg/mL) were dissolved in PBS buffer, as stock solutions. Before using them on cell lines, they were diluted to appropriate concentrations in the culture medium. All the drugs and solvents used here were purchased from Sigma-Aldrich (Sigma-Aldrich, St. Louis, MO, USA).

### 2.3. MTT Assay

The impact of carbidopa (0.2–20 µM), rasagiline (2–200 µg/mL), MTX (0.01–10 µM), and CDDP (0.1–10 µg/mL) [29,30] on cell viability was assessed in the MTT assay. The FM55P, FM55M2, A375, and SK-MEL28 cell lines (density: 2–3 × 10^4^ cells/mL) and the HaCaT cell line—a model of human keratinocyte cells [32] (density: 1 × 10^4^ cells/mL)—were plated on microtiter plates (NEST Biotechnology, Wuxi, China). After incubating the cell lines for 24 h, the medium was replaced with fresh medium, to which increasing concentrations of carbidopa, rasagiline, MTX, and CDDP were added. The MTT test was performed after 72 h; the subsequent steps of the procedure were described in previous publications [29,30,31]. In the MTT assay, the percentage of inhibition of cell viability along with the concentrations of rasagiline, carbidopa, MTXb and CDDP were determined as described in the protocol published earlier [30]. The median inhibitory concentration (IC_50_ value) for each tested drug was determined with a computer-assisted log-probit method [33]. Next, the combinations of rasagiline + MTX and rasagiline + CDDP (both in a fixed-ratio of 1:1) were tested in the FM55P, FM55M2, A375, and SK-MEL28 cell lines, from which the experimentally derived IC_50 mix_ values for the two-drug mixtures that were determined in the MTT assay. Each experiment was performed in triplicate.

### 2.4. LDH Test

The cytotoxicity of rasagiline was assessed in the LDH assay (Cytotoxicity Detection Kit PLUS LDH, Roche Diagnostics, Mannheim, Germany), in which the lactate dehydrogenase activity (released from damaged cells into the medium) was measured. The LDH assay in the FM55P, FM55M2, A375, SK-MEL28, and HaCaT cell lines was carried out after 72 h of incubation with rasagiline (at concentrations of 2–200 µg/mL), according to the manufacturer’s instructions. Positive control (ctr+) was the maximum LDH release, and it was achieved through addition of Lysis buffer (Triton x-100; Sigma-Aldrich, St. Louis, MO, USA) to untreated control cells, as described elsewhere [29,30,31].

### 2.5. BrdU Test

The BrdU assay, assessing the effect of the test compound on proliferation, was assessed using an ELISA BrdU Kit from Roche Diagnostics (Mannheim, Germany). The assay was initiated similarly to the MTT and LDH assays. Cells were treated with increasing concentrations of rasagiline, except that after 48 h, 10 µL/well BrdU Labeling Solution (100 µM) was added. The subsequent test steps were performed according to the instructions provided by the test manufacturer and described elsewhere [29,30,31].

### 2.6. Flow Cytometry—Cell-Cycle Analysis

Tested cells were seeded on 6-well microplates (density mentioned above; 2 mL/well). On the following day, the culture medium was removed, and the cells were exposed for 72 h to rasagiline at its IC_50_ concentrations. Untreated cells were used as a control. Next, cells were fixed with 80% ethanol at −20 °C for 2 h, then stained with PI/RNase Staining Buffer (Becton Dickinson, San Jose, CA, USA) according to the manufacturer’s protocol. The cells were analyzed on the BD Accuri C6 Plus flow cytometer (Becton Dickinson, San Jose, CA, USA).

### 2.7. Isobolographic Analysis

Pharmacodynamic interactions between rasagiline and two drugs (MTX and CDDP) in tested melanoma cell lines were classified isobolographically in the MTT assay. The parallelism of two concentration–response lines was verified for two combinations: rasagiline + MTX and rasagiline + CDDP, as previously described [33,34,35]. From the IC_50_ values of rasagiline, MTX, and CDDP (when administered alone in the MTT assay), the median additive inhibitory concentrations for the mixtures (IC_50 add_) (at a fixed ratio of 1:1) of rasagiline + MTX or rasagiline + CDDP were calculated, as described previously [29,34].

### 2.8. Statistical Analysis

Data from the MTT, LDH, and BrdU assays were first verified for their normal Gaussian distribution by means of the post-hoc Shapiro–Wilk normality test and Kolmogorov–Smirnov test. Subsequently, the data with confirmed normal distribution were analyzed with a one-way ANOVA test, followed by the Dunnett’s multiple comparisons post-hoc test, and every possible comparison between the study groups was considered. The IC_50_ values (for rasagiline + MTX and rasagiline + CDDP) from the isobolographic analysis were statistically compared with the unpaired Student’s *t*-test with Welch’s correction, as suggested earlier [36]. All statistical tests were calculated with version 8.0 of GraphPad Prism (San Diego, CA, USA).

## 3. Results

### 3.1. Effects of Rasagiline on Malignant Melanoma Cell Viability in the MTT Assay

Rasagiline inhibited (in a concentration-dependent manner) the viability of the four malignant melanoma cell lines (Figure 1a–d) and normal human keratinocytes (Figure 1e).

The cell viability in four malignant melanoma cell lines was significantly inhibited when rasagiline was administered at concentrations ranging from 2 µg/mL to 80–100 µg/mL (Figure 1a–d). Similarly, rasagiline had an impact on the viability of normal keratinocytes; however, at the highest tested concentration of 200 µg/mL, rasagiline inhibited cell viability to approximately 65%, *p* < 0.0001 (Figure 1e). The IC_50_ values for rasagiline, CDDP, and MTX in FM55P, FM55M2, A375, and SK-MEL28 cell lines are presented in Table 1. Of note, rasagiline in this study was dissolved a stock solution and tested on various melanoma cell lines, taking into account its mass concentrations of the solution (µg/mL). However, to standardize the unit system for rasagiline, its IC_50_ values were also expressed in molar units (µM) (Table 1). This was the main reason to report the IC_50_ values for rasagiline in both units (µg/mL and µM). Notably, the IC_50_ values for CDDP and MTX were determined experimentally in our previous studies [29,30]. The experimentally derived IC_50_ value for rasagiline in the HaCaT cell line was determined in approximation, using the log-probit method, amounting to 884.96 µg/mL (3310.2 µM).

Carbidopa at concentrations up to 20 μM weakly inhibited the viability of melanoma cells in the MTT assay. Notably, the maximal concentration of carbidopa tested in the cell line experiments was 20 μM because it was the highest possible concentration obtained experimentally due to the solubility of carbidopa in DMSO and the final safe concentration of 0.1% DMSO used in the culture medium. Carbidopa at a concentration of 20 µM reduced melanoma cell proliferation by only a few percent (i.e., maximally by 13.16% in the case of the FM55P cell line, *p* < 0.0001) (Figure 2a–d). This was the main reason not to investigate its anti-proliferative effects in further experiments involving drug combinations.

The experimentally determined IC_50_ value for rasagiline in the HaCaT (a normal human keratinocyte) cell line allowed for the selectivity index for rasagiline to be calculated in various melanoma cell lines (A375, SK-MEL28, FM55P, and FM55M2). The selectivity index is a quotient of the IC_50_ values for malignant melanoma cell lines and the IC_50_ for HaCaT, as determined in the MTT assay. In this study, selectivity index for rasagiline ranged from 8.22 to 28.18 for melanoma cell lines (A375, SK-MEL28, FM55P, and FM55M2) (Figure 3).

### 3.2. Effects of Rasagiline on Malignant Melanoma Cell Cytotoxicity in the LDH Assay

Rasagiline (200 µg/mL) produced significant LDH leakage in the FM55P and FM55M2 cell lines (Figure 4a,b; *p* < 0.0001 and *p* < 0.001 respectively). In the case of A375, SK-MEL28, and HaCaT cell lines, rasagiline was not cytotoxic to the studied cells at concentrations up to 200 µg/mL (Figure 4c–e).

### 3.3. Effects of Rasagiline on Malignant Melanoma Cell Proliferation in the BrdU Assay

Rasagiline showed an effect on the proliferation of all the tested cell lines, causing a dose-dependent decrease in DNA synthesis (Figure 5). The BrdU assay assesses cellular proliferation by assessing the binding of BrdU (5-bromo-2′-deoxyuridine) to cellular DNA in cells. Only the highest concentration of rasagiline (200 µg/mL) caused slight inhibition of cell proliferation of normal keratinocytes (Figure 5e). In the case of melanoma cells, a statistically significant inhibition of cell proliferation was observed, from a concentration of 150 µg/mL for FM55M2 and SK-MEL28 lines (Figure 5b,d) and 200 µg/mL for FM55P and A375 lines (Figure 5a,c).

The results of this experiment emphasize that rasagiline affects cell proliferation only at high concentrations, whereas at low concentrations, it affects their metabolic activity, as demonstrated by the MTT assay.

### 3.4. Effects of Rasagiline on Malignant Melanoma Cell Cycle Progression

In the FM55P cell line (Figure 6a), after rasagiline IC_50_ treatment, a significant increase in the number of cells in the G0/G1 phase was observed, reaching an average of 76.48% ± 0.34. This was accompanied by a decrease in the percentage of cells in the S phase (average 7.67% ± 0.36), which may indicate inhibition of DNA replication and arrest of the cell cycle before entering the synthesis phase. The percentage of cells in the G2/M phase did not change significantly and averaged 9.91% ± 0.56. These results suggest that rasagiline may affect the cell cycle of FM55P cells by inducing arrest in the G0/G1 phase and limiting cell progression to the S phase. This effect may indicate a potential cytostatic or antiproliferative effect of rasagiline on the tested cells. In the case of the A375 cell line (Figure 6c), rasagiline treatment resulted in a significant shift of the cell cycle towards the G0/G1 phase—on average, 75.46% ± 1.62. At the same time, a significant reduction in the percentage of cells in the S phase was observed, to 10.83% ± 1.20, which may indicate a reduction in DNA synthesis. The percentage of cells in the G2/M phase also decreased slightly and amounted to an average of 18.93% ± 0.74. This effect may indicate potential cytostatic properties of rasagiline against FM55P and A375 cancer cells.

In both metastatic melanoma cell lines FM55M2 (Figure 6b) and SK-MEL28 (Figure 6d), no significant shifts in cell cycle distribution were observed after rasagiline treatment. Rasagiline (at the IC_50_ concentration) neither significantly affected cell cycle arrest nor induced proliferative changes in FM55M2 and SK-MEL28 cells. Therefore, it can be assumed that the mechanism of action of rasagiline in these cell lines does not involve modulation of cell cycle checkpoints, at least under the time-concentration conditions tested.

### 3.5. Isobolographic Interactions Between Rasagiline and Cisplatin and Mitoxantrone

The linearly related log-probit concentration–response inhibitory effects of rasagiline were tested for the parallelism with the log-probit concentration–response inhibitory effects of cisplatin (CDDP) and mitoxantrone (MTX). The concentration–response line of rasagiline was parallel to that of CDDP in three tested cell lines: FM55P, A375, and SK-MEL28 (Table 2; Appendix A). In the case of the combination of rasagiline and MTX, their concentration–response lines were mutually parallel for the two cell lines: FM55P and SK-MEL28 (Table 2; Appendix A). Notably, the parallelism of log-probit lines is crucial for the isobolographic analysis of interactions and the definition of additivity for both, parallel and nonparallel concentration–response lines [34,35,36,37].

Isobolographically, the combination of rasagiline with CDDP (at a fixed ratio of 1:1) exerted antagonistic interactions (* *p* < 0.05) in both the A375 and SK-MEL28 cell lines (Figure 7c,d), and additive interactions with a tendency toward antagonism in the FM55P and FM55M2 cell lines (Figure 7a,b).

Synergistic interaction (* *p* < 0.05) was observed isobolographically between rasagiline and MTX (at the fixed ratio of 1:1) in the FM55P melanoma cell line (Figure 8a). The combination of rasagiline with MTX had additive effects with a tendency toward synergy on the remaining FM55M2, A375, and SK-MEL28 cell lines (Table 2 and Table 3, Figure 8b–d).

## 4. Discussion

In this study, carbidopa has a negligible anti-viability effect, inhibiting the proliferation of malignant melanoma cell lines by only several percent at a maximal tested concentration of 20 µM. In vitro experiments with carbidopa, conducted on melanoma (A375) and breast cancer (MCF-7) cell lines, showed that 15 µM carbidopa did not affect the viability of melanoma in the A375 cell line, but increased the proliferation of breast cancer cells. To explain this phenomenon, it should be borne in mind that carbidopa may cause a modification of tryptophan metabolism and the emergence of a new metabolite indole-3-acetonitrile (IAN), which promoted a concentration-dependent increase in viability of both cell lines. The IAN metabolite is a potential contributor to increased viability of tumor cells, which may suggest why carbidopa is ineffective in reducing the incidence of breast cancer and melanoma in PD patients [38]. Additionally, carbidopa inhibited the proliferation of pancreatic cancer cells both in vitro (inhibiting the viability of BxPC-3 and Capan-2 cells) and in vivo [39]. Carbidopa also reduced the growth of prostate cancer [40]. Experimental evidence suggested that carbidopa (tested at concentrations of 10–2000 µM) strongly inhibited the viability of B16(F10) murine melanoma cells. Carbidopa at a concentration of 250 µM was found to be lethal to cells after 48 h of incubation [41]. However, special attention should be paid to the latter experiments because of the solubility limit of 20 µM carbidopa in 0.1% DMSO used in the cell line in vitro experiments. Perhaps other solvents allowed carbidopa to be dissolved at concentrations of 250 and 2000 µM [40], but this finding should be confirmed in more advanced studies in relation to the direct toxicity of the solvents to exposed cells in culture. Other researchers found that 4-S-cysteaminylphenol (4-S-CAP, a melanin-like pigment-forming tyrosinase substrate) significantly inhibited the growth of B16 melanoma cells inoculated into C57BL/6J mice. The administration of a combination of L-DOPA and carbidopa to mice significantly increased the anti-melanoma effect of 4-S-CAP [42]. Interestingly, human studies have shown that orally administered L-DOPA/carbidopa is ineffective in the treatment of advanced melanoma when maximum tolerated doses (up to 4 g per day) are used, as demonstrated in a group of 17 patients with metastatic melanoma [43].

Rasagiline (a selective irreversible MAO-B inhibitor) was approved in Europe in 2005 and in the United States in 2006 for the treatment of PD. Several patients were diagnosed with melanoma during clinical trials of rasagiline; therefore, special attention is advised to monitor patients taking rasagiline [25]. Our results demonstrated that rasagiline inhibits human melanoma cell viability in a concentration-dependent manner. Other studies have shown that rasagiline can prevent dexamethasone-induced brain cell death, as demonstrated in human SH-SY5Y neuroblastoma cells and 1242-MG glioblastoma cells [43]. Rasagiline showed greater inhibition of MAO-B enzymatic activity and prevention of DNA damage (TUNEL assay) than did selegiline and 1-R-aminoindan [44]. Rasagiline inhibited the reduction in mitochondrial membrane potential, cytochrome c release and apoptosis induced by N-methyl(R)salsolinol in SH-SY5Y (neuroblastoma) cells, which proves that rasagiline also has a direct effect on mitochondria [45]. Both in vivo and in vitro experiments showed that rasagiline increased the levels of Bcl-2 and neurotrophic factors; thus, rasagiline protected neurons against cell death in animal and cellular models of neurodegeneration [45,46]. Rasagiline directly suppresses mitochondrial apoptotic signaling [47] and induces anti-apoptotic Bcl-2 and pro-survival neurotrophic factors [48].

Our experiments confirmed that rasagiline inhibited the viability of four different human malignant melanoma cell lines at a concentration of 2 µg/mL, and the IC_50_ of rasagiline mesylate ranged from 31.4 µg/mL (≈117 µM) to 107.71 µg/mL (≈403 µM), depending on the tested cell line. Interestingly, the results of our experiments are consistent with those reported by Meier-Davis et al., who tested rasagiline solutions administered orally (1.5 mg/mL of rasagiline, administered at a dose of 15 mg/kg body weight) or in the form of patches transdermally (10% rasagiline) to female nude mice harboring the SK-MEL28 human melanoma cell line. Compared with placebo, oral administration of rasagiline reduced the tumor volume by 22% and transdermal administration by 23% (one patch) and 37% (two patches), respectively. The dose of rasagiline used (15 mg/kg body weight) is approximately one thousand times greater than the clinical dose used in humans (up to 1 mg/day), yet rasagiline reduced melanoma tumor growth during treatment [49].

The selectivity index for rasagiline calculated in this study for A375, FM55P, FM55M2, and SK-MEL28 cell lines (in comparison to the normal human keratinocyte HaCaT cell line) confirmed that rasagiline possesses the ideal anti-melanoma profile, i.e., low toxic selectivity to normal healthy cells and high selective toxicity to malignant melanoma cells. Of note, the anticancer drugs with a selectivity index value higher than 10 are considered to be safe for normal healthy cells with specific preferential toxicity to cancerous cells [50]. Previously, it has been documented that AM1172—a hydrolysis-resistant endocannabinoid analog—had unfavorable selectivity index (less than 0.2) in various malignant melanoma cell lines as compared to the normal healthy keratinocyte HaCaT cell line [51].

The results presented herein have certain limitations related to the fact that the results concern only in vitro studies. The potentially best synergistic combination of rasagiline with MTX would be worthy of testing in in vivo animal studies. Translation of such a synergistic interaction from the in vitro study to in vivo conditions would be of crucial importance for elaborating effective pharmacotherapy. In the case of the combination of rasagiline with CDDP, an antagonistic interaction demonstrated herein, votes against its clinical application due to the abolition of the anticancer effects. Due to the interdependence of both Parkinson’s disease and melanoma, it would be worth taking a closer look at the drugs used together to detect any undesirable pharmacodynamic interactions. On the other hand, one future research direction may be biomolecular studies using both cellular and animal models to reveal any correlations (at genetic and molecular levels) between rasagiline and human melanoma. The anti-viability effects of rasagiline reported herein in human melanoma cell lines can shed more light on mechanisms related with tumorigenesis in melanocytes to help develop some effective therapeutic options in the future.

## 5. Conclusions

Summing up, the combination of rasagiline with CDDP was characterized by an unfavorable antagonistic interaction (or additivity with a tendency toward antagonism) in various malignant melanoma cell lines. In contrast, the combination of rasagiline with MTX produced a beneficial synergistic interaction or additive effect with a tendency toward synergy in various malignant melanoma cell lines. This may be due to the different mechanisms of action of the two chemotherapeutic drugs. Further, in vivo and in vitro experiments should shed more light on rasagiline and its combinations with other chemotherapeutics since rasagiline has the potential to become an anti-melanoma drug, especially when combined as an add-on therapy. Furthermore, multi-drug therapies are sought because they can offer significantly better therapeutic effects than monotherapy (reduction of side effects and lack of emerging treatment resistance in the case of monotherapy).

## Figures and Tables

**Figure 1 cancers-17-02563-f001:**
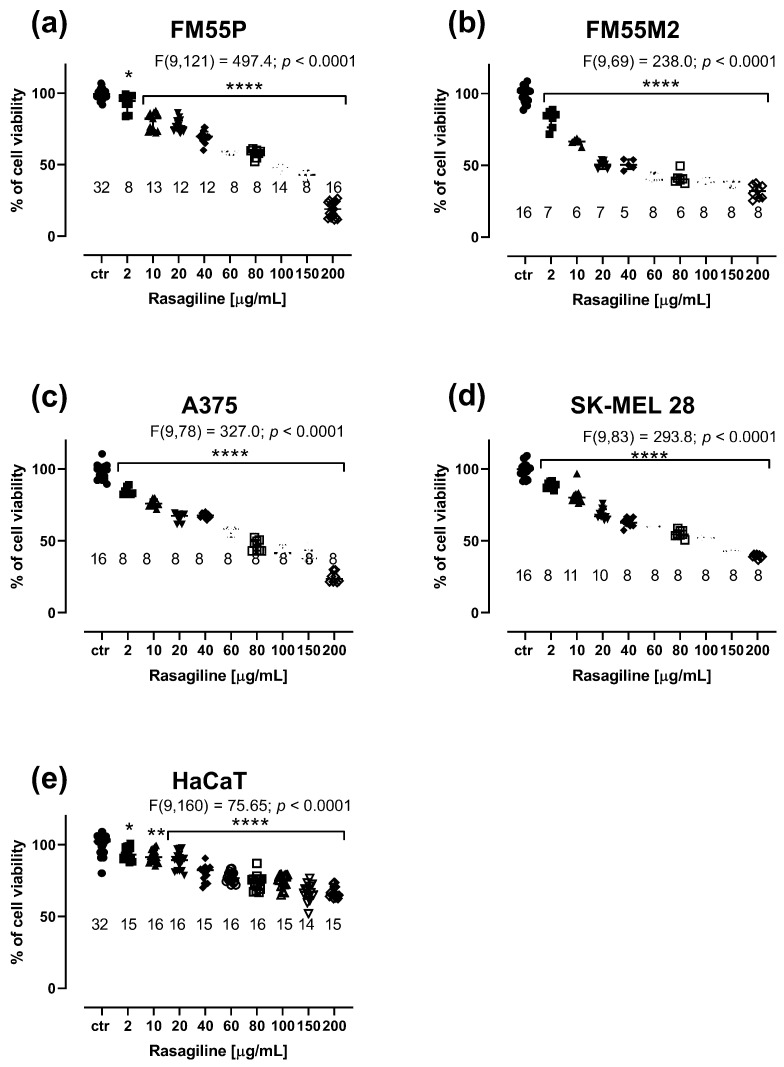
Influence of rasagiline on the viability of malignant melanoma cell lines: FM55P (**a**), FM55M2 (**b**), A375 (**c**), SK-MEL28 (**d**), and the normal human keratinocyte HaCaT (**e**) in the MTT assay. Columns represent the means ± SEMs. N values for each group have been placed on the graph, but each group was tested in triplicate. * *p* < 0.05, ** *p* < 0.01, and **** *p* < 0.0001 vs. the control (ctr) group (one-way ANOVA followed by the Dunnett’s post-hoc test).

**Figure 2 cancers-17-02563-f002:**
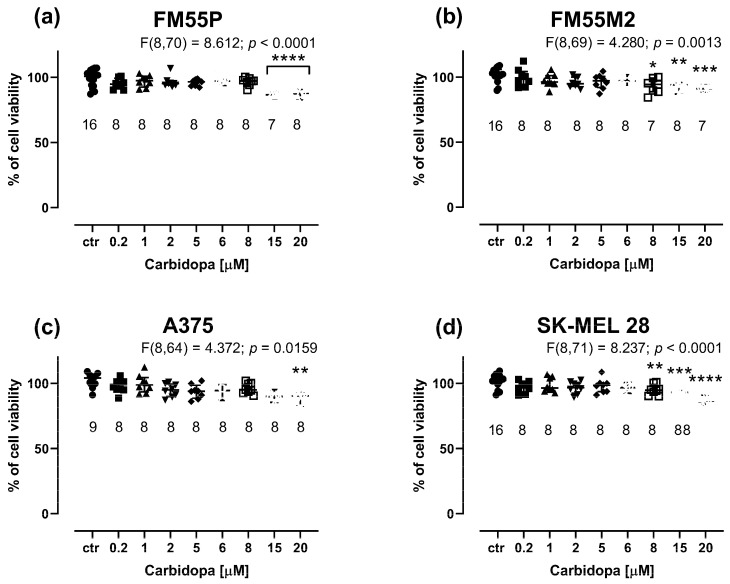
Influence of carbidopa on the viability of malignant melanoma cell lines: FM55P (**a**), FM55M2 (**b**), A375 (**c**), and SK-MEL28 (**d**) in the MTT assay. The columns represent the means ± SEMs. N values for each group have been placed on the graph, but each group was tested in triplicate. * *p* < 0.05, ** *p* < 0.01, *** *p* < 0.001, and **** *p* < 0.0001 vs. the control (ctr) group (one-way ANOVA followed by the Dunnett’s post-hoc test).

**Figure 3 cancers-17-02563-f003:**
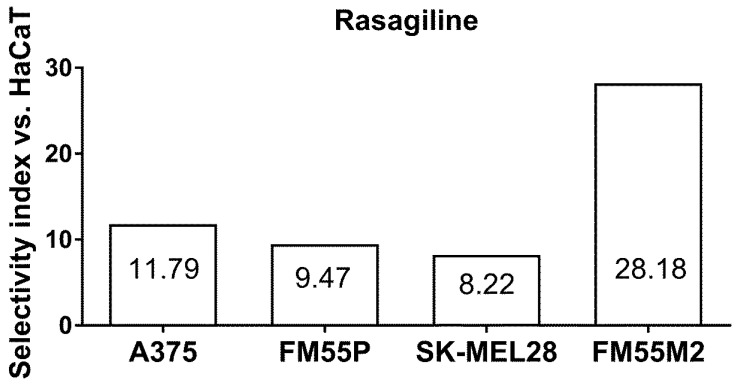
Selectivity index for rasagiline in the MTT assay. Columns represent the selectivity index of rasagiline (as a quotient of its IC_50_ values determined in the MTT assay for HaCaT and malignant melanoma cell lines).

**Figure 4 cancers-17-02563-f004:**
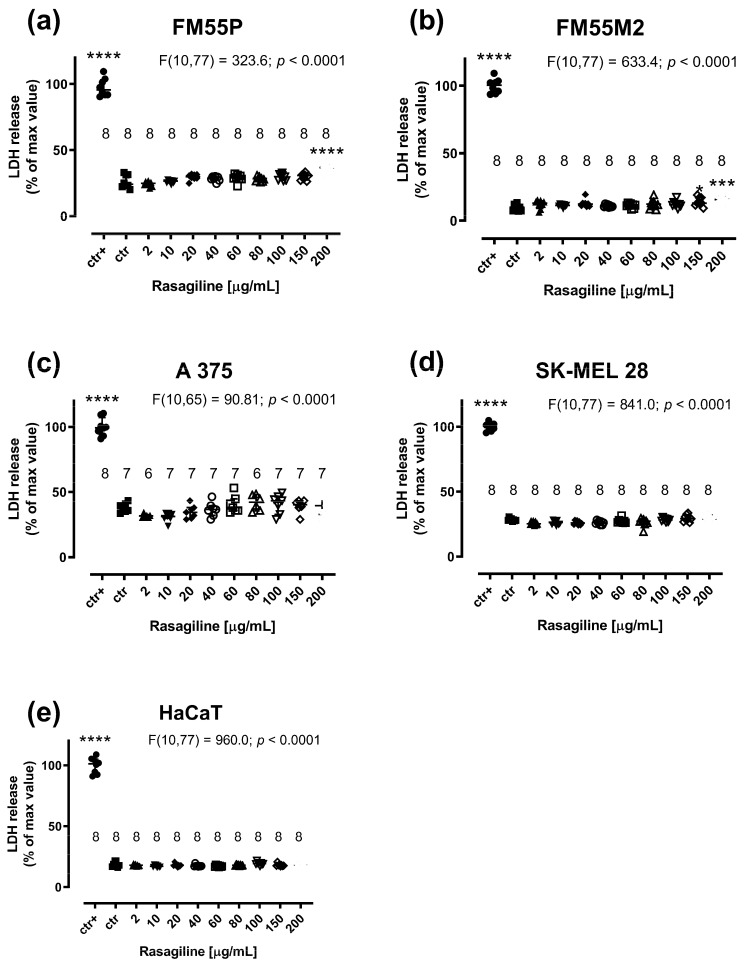
Influence of rasagiline on cytotoxicity in malignant melanoma cell lines: FM55P (**a**), FM55M2 (**b**), A375 (**c**), SK-MEL28 (**d**), and normal human keratinocytes HaCaT (**e**) in the LDH assay. Columns represent the means ± SEMs. N values for each group have been placed on the graph, but each group was tested in triplicate. * *p* < 0.05, *** *p* < 0.001 and **** *p* < 0.0001 vs. the control (ctr) group; ctr+—cells treated with lysis buffer (one-way ANOVA followed by the Dunnett’s post-hoc test).

**Figure 5 cancers-17-02563-f005:**
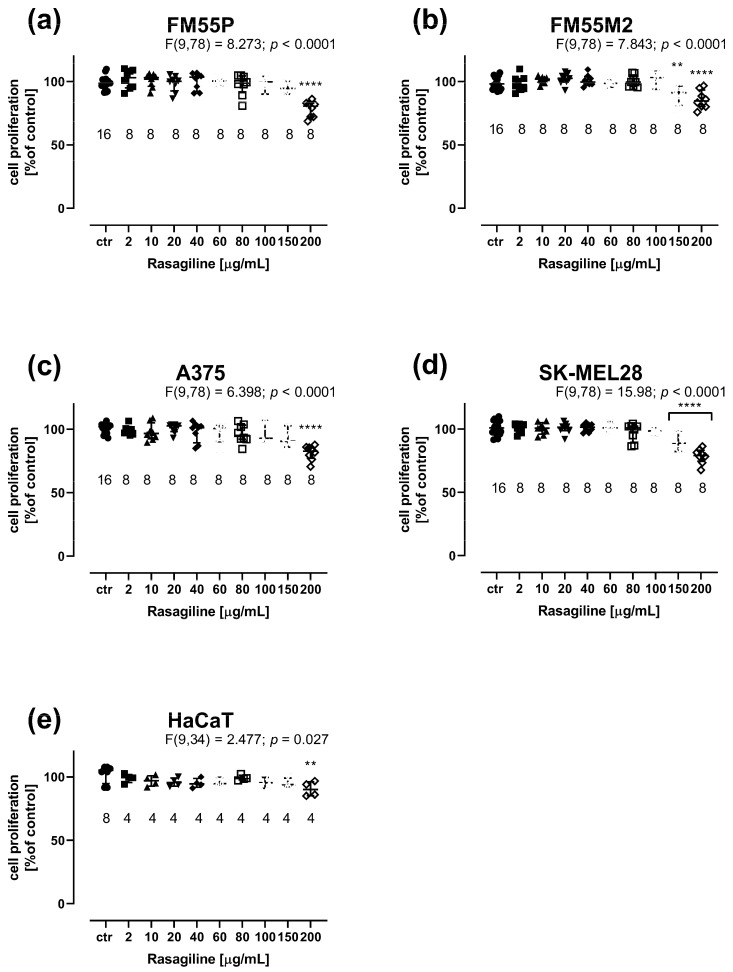
Influence of rasagiline on cell proliferation in malignant melanoma cell lines: FM55P (**a**), FM55M2 (**b**), A375 (**c**), SK-MEL28 (**d**), and normal human keratinocytes HaCaT (**e**) in the BrdU assay. Columns represent the means ± SEMs. N values for each group have been placed on the graph, but each group was tested in triplicate. ** *p* < 0.01 and **** *p* < 0.0001 vs. the control (ctr) group (one-way ANOVA followed by the Dunnett’s post-hoc test).

**Figure 6 cancers-17-02563-f006:**
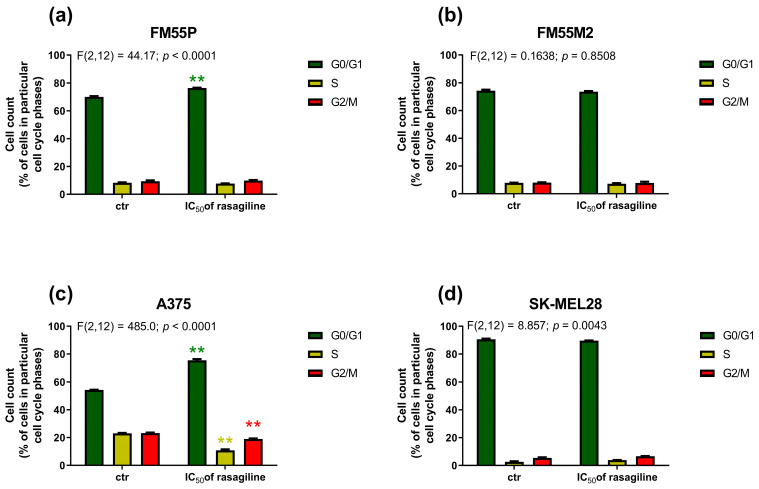
Effects of rasagiline on cell cycle progression in four human melanoma cell lines: FM55P (**a**); FM55M2 (**b**), A375 (**c**), and SK-MEL28 (**d**). Cells were exposed for 72 h to culture medium alone (control) or culture medium containing the IC_50_ concentrations of rasagiline. DNA content was determined using flow cytometry after cell staining with propidium iodide. Analysis was used to define the percentage of cells in G0/G1, S and G2-M cell cycle phases. Data represent the mean ± SEMs of three independent trials, ** *p* < 0.01 vs. control (one-way ANOVA test with Dunnett’s post-hoc test).

**Figure 7 cancers-17-02563-f007:**
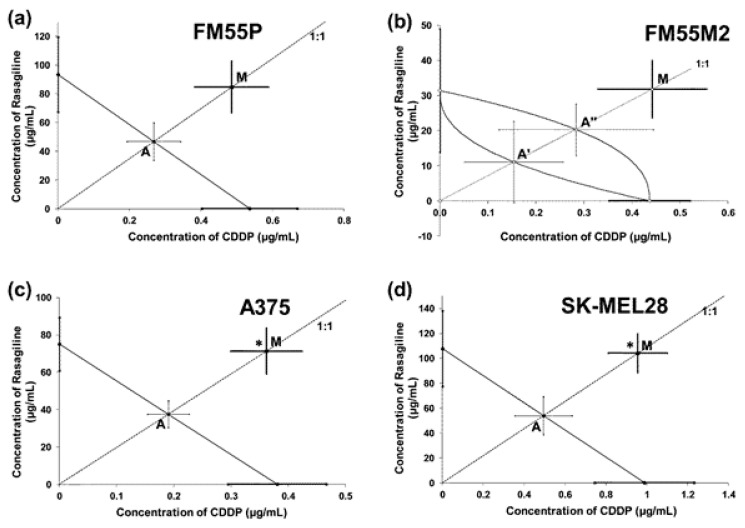
Interactions of rasagiline with cisplatin (CDDP) in the FM55P (**a**), FM55M2 (**b**), A375 (**c**), and SK-MEL28 (**d**) cell lines in the MTT assay. The IC_50_ (±SEM) values for rasagiline and CDDP were plotted in the Cartesian system of coordination. The theoretically additive IC_50add_ values were plotted graphically as the points A or A’, A’’, whereas the experimentally derived IC_50mix_ values were plotted as the point M on each multi-part graph. * *p* < 0.05 vs. the respective IC_50_ value (Student’s *t*-test with Welch’s correction).

**Figure 8 cancers-17-02563-f008:**
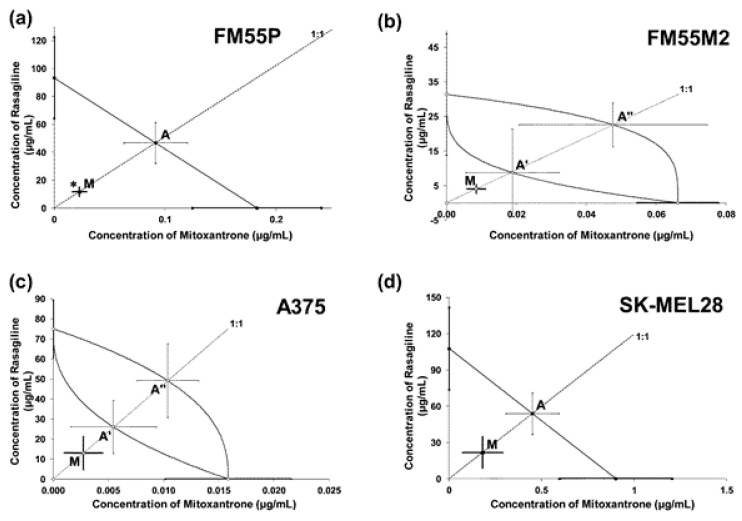
Interactions of rasagiline with mitoxantrone in the FM55P (**a**), FM55M2 (**b**), A375 (**c**), and SK-MEL28 (**d**) cell lines in the MTT assay. The IC_50_ (±SEM) values for rasagiline and mitoxantrone were plotted in a Cartesian system of coordination. The theoretically additive IC_50add_ values were plotted graphically as the points A or A’, A’’, whereas the experimentally derived IC_50mix_ values are plotted as the point M on each multi-part graph. * *p* < 0.05 vs. the respective IC_50_ value (Student’s *t*-test with Welch’s correction).

**Table 1 cancers-17-02563-t001:** The anti-proliferative effects of rasagiline, cisplatin, and mitoxantrone on four various malignant melanoma cell lines in the MTT assay.

Drug/Cell Line	FM55P	FM55M2	A375	SK-MEL28	References
Rasagiline mesylate	349.44 ± 97.52 µM	117.45 ± 56.89 µM	280.69 ± 53.30 µM	402.89 ± 113.83 µM	this study
Rasagiline mesylate	93.42 ± 26.07 µg/mL	31.4 ± 15.21 µg/mL	75.04 ± 14.25 µg/mL	107.71 ± 30.43 µg/mL	this study
Cisplatin	1.49 ± 0.30 μM	1.70 ± 0.35 μM	1.29 ± 0.34 μM	3.30 ± 0.70 μM	[29,30]
Mitoxantrone	0.35 ± 0.10 μM	0.13 ± 0.02 μM	0.04 ± 0.02 μM	1.74 ± 0.51 μM	[30]

The results are presented as IC_50_ values (means ± SEMs).

**Table 2 cancers-17-02563-t002:** Isobolographic analysis of interactions of rasagiline with CDDP and MTX (at the fixed ratio of 1:1) for parallel concentration–response effects in various malignant melanoma cell lines in the MTT assay.

Drug Combination	Cell Line	IC_50mix_ (n_mix_) [µg/mL]	IC_50add_ (n_add_) [µg/mL]	t Statistics	Interaction
Rasagiline+CDDP	A375	71.66 ± 12.37 * (96)	37.71 ± 7.17 (212)	t_161_ = 2.374; *p* = 0.019	Antagonistic
	FM55P	85.15 ± 18.11 (96)	46.98 ± 13.10 (188)	t_193_ = 1.708; *p* = 0.089	Additive
	SK-MEL28	105.10 ± 15.84 * (96)	54.35 ± 15.34 (188)	t_247_ = 2.302; *p* = 0.022	Antagonistic
Rasagiline+MTX	FM55P	11.67 ± 3.28 * (96)	46.80 ± 14.61 (164)	t_179_ = 2.346; *p* = 0.020	Synergistic
	SK-MEL28	21.85 ± 13.08 (96)	54.31 ± 17.16 (164)	t_258_ = 1.504; *p* = 0.134	Additive

The IC_50_ values are presented as the means ± SEMs. The IC_50mix_—experimentally-derived IC_50_; n_mix_—number of wells for experimental mixture; IC_50add_—theoretically additive IC_50_; n_add_—number of wells calculated for the additive mixture. * *p* < 0.05 vs. the respective IC_50add_ value. Statistical analysis of data was performed with the unpaired Student’s *t*-test with Welch’s correction.

**Table 3 cancers-17-02563-t003:** Isobolographic analysis of interactions of rasagiline with CDDP and MTX (at the fixed ratio of 1:1) for non-parallel concentration–response effects in various malignant melanoma cell lines in the MTT assay.

Drug Combination	Cell Line	IC_50mix_ (n_mix_) [µg/mL]	Lower IC_50add_ (n_add_) [µg/mL]	Upper IC_50add_ (n_add_) [µg/mL]	t Statistics	Interaction
Rasagiline+CDDP	FM55M2	32.24 ± 8.32 (96)	11.22 ± 7.41 (182)	20.57 ± 7.47 (182)	t_231_ = 1.044; *p* = 0.298	Additive
Rasagiline+MTX	FM55M2	4.06 ± 1.27 (96)	8.75 ± 7.67 (182)	22.64 ± 8.33 (182)	t_191_ = 0.6033; *p* = 0.547	Additive
	A375	12.95 ± 8.29 (96)	26.02 ± 13.18 (242)	49.17 ± 15.33 (242)	t_336_ = 0.8394; *p* = 0.402	Additive

The IC_50_ values are presented as the means ± SEMs. The IC_50add_ values were calculated from the lower and upper isoboles of additivity. The n_mix_—total number of wells experimentally determined; n_add_—total number of wells calculated for the additive two-drug mixture. Statistical analysis of data was performed with the unpaired Student’s *t*-test with Welch’s correction.

## Data Availability

The raw data supporting the conclusions of this article will be made available by the authors on request.

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
