# Peer review of "Rasagiline Inhibits Human Melanoma Cell Viability and Interacts Synergistically with Mitoxantrone and Antagonistically with Cisplatin—In Vitro Isobolographic Studies"

_cancers, 2025, doi:10.3390/cancers17152563_

Round 1
Reviewer 1 Report (Previous Reviewer 4)
Comments and Suggestions for Authors
Paper titled (the Analysis of antiviability effects of rasagiline and carbidopa on various human malignant melanoma cell lines: an in vitro isobolographic study. ) by Authors Danuta Krasowska et al. is an in vitro study described the effect of rasagiline and carbidopa on viability of human melanoma cell lines. The methods used in carrying out the study are adequate but lacks approrpate references at some occasions. The results are documented and clarified also thanks to the attached figures, which clearly show anti viabilitye ffect of the drugs. The discussion appears clear enough. Many studies are cited.
Here, I am listing some recommendations for improving the topic. please find these comments & provide a point to point reply and highlight the changes in the file and indicate at what page & line we can follow every change.
1 - Title: ( Analysis of antiviability effects of rasagiline and carbidopa on various human malignant melanoma cell lines: an in vitro isobolographic study. ) is not fully informative , please give a clear message for the reader about the results
2 - Abstract must be amended by some numerical values for key findings from the study.
3 - Key words: melanoma; rasagiline; drug interaction; cisplatin; mitoxantrone; isobolographic analysis
can be amended (use human malignant melanoma) instead of melanoma
4 - No need for (drug interaction)
5 - Introduction: is brief and although introduces the items of the study well, did not explore the rational or novelty of the study. (which is too important as these drugs are little to be repurposed for a new use)
6 - AIm of the study should be clear and clarify what was the aim and how authors acheived it
7 - Methods section in general is too brief and plagiarized and lacks references MUST be extensively revised
8 - Experimental design : please give references and rationale for the selected doses of drugs
9 - Authors should give the source of chemicals, kits and antibodies completely and consistently (code, company, town, state and country) & version for software
10 . Statistical tests and sample sizes for each experiment should be explicitly mentioned in the methods and figure legends.
11 - Authors have to check the normality of distribution of the results by a suitable post hoc test (such as Shapiro-Wilk test or K-S test) before deciding to choose certain ANOVA. If the normality test indicated normal dist of the data, so use one-way ANOVA, if not, use non parametric ANOVA. In all cases choose a suitable post-hoc test
12 - Authors should confirm in methods that "every possible comparison between the study groups was considered" and apply this in results
13 - Mention "n" in each illustration individually
14 - Use appropriate abbreviations for minutes, seconds...etc
15 - Every abbreviation in figures should be explained in the figure legend to be self-explanatory & stands alone.
16 - Ensure every abbreviation is explained at the first appearance in abstract & then in the body text
17 - Methods in general lacks references
18- Results are will described not in the best way and needs amendement to be explained
19- Figure resolutions are weak Especially Figure 1 ,2 ,3 ,4
20 -Please write the clinical potential of this paper in aim and conclusion
21 - write the limitations of this study and future directions after which
With the above revisions, I believe your manuscript will make a valuable contribution to the field. I encourage you to address these suggestions to improve the clarity and overall impact of your paper.
Author Response
Paper titled (the Analysis of antiviability effects of rasagiline and carbidopa on various human malignant melanoma cell lines: an in vitro isobolographic study. ) by Authors Danuta Krasowska et al. is an in vitro study described the effect of rasagiline and carbidopa on viability of human melanoma cell lines. The methods used in carrying out the study are adequate but lacks approrpate references at some occasions. The results are documented and clarified also thanks to the attached figures, which clearly show anti viabilitye ffect of the drugs. The discussion appears clear enough. Many studies are cited.
Here, I am listing some recommendations for improving the topic. please find these comments & provide a point to point reply and highlight the changes in the file and indicate at what page & line we can follow every change.
A: Thank You for reviewing our manuscript. We are grateful for your valuable comments.
1 - Title: ( Analysis of antiviability effects of rasagiline and carbidopa on various human malignant melanoma cell lines: an in vitro isobolographic study. ) is not fully informative , please give a clear message for the reader about the results
A: The authors thank you very much for your comment. We have proposed a new title:
„Rasagiline inhibits human melanoma cell viability and interacts synergistically with mitoxantrone and antagonistically with cisplatin – in vitro isobolographic studies”.
2 - Abstract must be amended by some numerical values for key findings from the study.
A: Thank you very much for this comment. We would like to note that the abstract provides IC50 values for rasagiline for all the tested cell lines and selectivity index values. The authors supplemented the data with increases and decreases in cell numbers in specific cell cycle phases, consistent with the results obtained from flow cytometry studies (lines 31-33).
3 - Key words: melanoma; rasagiline; drug interaction; cisplatin; mitoxantrone; isobolographic analysis
can be amended (use human malignant melanoma) instead of melanoma
A: Thank you for this comment. The authors have added "human malignant melanoma" to key words. Line 43
4 - No need for (drug interaction)
A: Thank you for this comment. The authors have removed „drug interaction” from Key words (Line 43)
5 - Introduction: is brief and although introduces the items of the study well, did not explore the rational or novelty of the study. (which is too important as these drugs are little to be repurposed for a new use)
A: In lines 94-97 and 103-105, the authors added important information regarding the innovative approach to the study. Primarily, the performance of cytometric studies and the search for two-drug combinations that are significantly superior from a clinical perspective (no drug resistance over time).
6 - AIm of the study should be clear and clarify what was the aim and how authors acheived it
A: Thank you for your comment. The broad purpose of the research was established in the final paragraph of the introduction (starting at line 91). It has now been supplemented with relevant considerations (as outlined above).
7 - Methods section in general is too brief and plagiarized and lacks references MUST be extensively revised
A: The Materials and Methods section has been significantly expanded. The authors have added the stocks used and the tested dosages (lines 130-131 and 138-139). A detailed MTT assay protocol has been added in lines 144-151, and the LDH assay has been added in lines 166-171.
8 - Experimental design : please give references and rationale for the selected doses of drugs
A: As mentioned above, the authors added the stocks used and the tested dosages (lines 130-131 and 138-139). The stocks used were verified in the material safety data sheets and their solubility data according to the PubChem database.
9 - Authors should give the source of chemicals, kits and antibodies completely and consistently (code, company, town, state and country) & version for software
A: Thank you for your comment. For each reagent or kit used, the company name and all necessary data are provided (including lines 110, 111, 116, 142, 150, 161, and 176). For GraphPad Prism, line 219 states that version 8.0 was used.
10 . Statistical tests and sample sizes for each experiment should be explicitly mentioned in the methods and figure legends.
A: We are very grateful for this suggestion. The statistical analysis section in the Materials and Methods chapter (lines 212-216) describes in detail the analysis performed for the MTT, LDH, and BrdU assays. Furthermore, all graphs have been modified as indicated, with appropriate legend entries and n numbers added for each column in the graph, as requested.
11 - Authors have to check the normality of distribution of the results by a suitable post hoc test (such as Shapiro-Wilk test or K-S test) before deciding to choose certain ANOVA. If the normality test indicated normal dist of the data, so use one-way ANOVA, if not, use non parametric ANOVA. In all cases choose a suitable post-hoc test
A: Thank you very much for this suggestion. Of course, we performed tests such as the Shapiro-Wilk test or K-S test, as detailed in the Materials and Methods chapter, in the section on statistical analysis (lines 212-216).
12 - Authors should confirm in methods that "every possible comparison between the study groups was considered" and apply this in results
A: As mentioned earlier, the description of the statistical methods used in the study was reworded (lines 212-216). We have added the mentioned sentence as requested. Additionally, some important entries were added in the legends under the corrected figures.
13 - Mention "n" in each illustration individually
A: Thank you for your comment. As mentioned earlier, for the figures from the MTT, LDH, and BrdU tests, n numbers were added for each column in the graph.
14 - Use appropriate abbreviations for minutes, seconds...etc
A: The abbreviation for minutes (min.) has been used throughout the manuscript (lines 169, 183,185).
15 - Every abbreviation in figures should be explained in the figure legend to be self-explanatory & stands alone.
A: Thank you for your comment. All figures have been revised to take into account the Reviewer's comments.
16 - Ensure every abbreviation is explained at the first appearance in abstract & then in the body text
A: Thank you for your comment. The text has been re-checked, and abbreviations are explained when they first appear in both the abstract and the text (e.g., on lines 23-24 or 47, 75, 101). Added explanation of abbreviations (line 62-64, 75). Additionally, a list of abbreviations used has been added at the end of the article (line 510).
17 - Methods in general lacks references
A: As mentioned above The Materials and Methods section has been significantly expanded and references to our previous studies have been added as suggested.
18- Results are will described not in the best way and needs amendement to be explained
A: Thank you for your comment. The description of the results has been enriched with statistical significance notes for individual results (e.g., lines 236 and 257). The authors particularly focused on improving the graphs for better understanding of the results.
19- Figure resolutions are weak Especially Figure 1 ,2 ,3 ,4
A: Thank you very much for this valuable feedback. The figures included in the template are losing quality and becoming less legible. All figures have been included in the manuscript as separate, high-quality figures.
20 -Please write the clinical potential of this paper in aim and conclusion
A: The clinical potential is mentioned in the introduction (lines 103-105) and in the summary (lines 483-486).
21 - write the limitations of this study and future directions after which
A: The limitations of our study are described in the discussion summary paragraph (lines 459-468). Additionally, the authors have added a short paragraph with future directions (lines 468-474).
With the above revisions, I believe your manuscript will make a valuable contribution to the field. I encourage you to address these suggestions to improve the clarity and overall impact of your paper.
Reviewer 2 Report (Previous Reviewer 3)
Comments and Suggestions for Authors
The study by Krasowska D. et al was focused on the analysis of the anticancer effects of rasagiline and carbidopa, the well-known drugs against melanoma with slightly different mechanism of action: rasagiline is the inhibitor of monoaminooxidaseB (MAO-B) while decarboxylase DOPA is the target for carbidopa. The current manuscript is the second submission so it is reasonable to analyse the changes the authors made to improve the quality of the paper.
First of all the authors agreed with my (and probably also expressed by other reviewers) opinion that carbidopa has virtually no effect on viability of melanoma cell cultures used by the authors. In the new version the authors confirmed it right in the Abstract and though carbidopa remains in the title of the article all the data presented correspond to the effects of rasagiline. At the same time the authors did not forget to discuss the phenomenon of the lack of effect of carbidopa on the viability of melanoma cells which they associated with the changes induced by carbidopa in tryptophan metabolism and the appearance of IAN.
Overally, it can be said that the article has been significantly revised and supplemented with several significant results based on BrdU-test and flow-cytometry to assess cell proliferation and analyse cell cycle stages being arrested. My comments were also taken into account. For instance, in the legend to Fig.3 the term selectivity index is explained more carefully “a quotient of IC50 values for malignant melanoma cell lines and IC50 for HaCaT,determined in the MTT assay” like I have reccomended. The lysing agent in the lysis buffer used to determine a 100% level of LDH in cells (Fig.4) is indicated as TX-100 in “LDH test” subsection of Materials and Methods. My opinion is that now the article appears to be far more interesting. A new mathematical approach, isobolographic analysis of drugs interaction makes it possible for the authors to distinguish synergistic, additive and antagonistic drug interactions and make important recommendations to cancer clinicians regarding the use of rasagiline together with MTX which exerted the most desirable synergistic effect on melanoma cell proliferation and should not use rasagiline in combination with CDDP. The article can be published in Cancer
Author Response
The study by Krasowska D. et al was focused on the analysis of the anticancer effects of rasagiline and carbidopa, the well-known drugs against melanoma with slightly different mechanism of action: rasagiline is the inhibitor of monoaminooxidaseB (MAO-B) while decarboxylase DOPA is the target for carbidopa. The current manuscript is the second submission so it is reasonable to analyse the changes the authors made to improve the quality of the paper.
First of all the authors agreed with my (and probably also expressed by other reviewers) opinion that carbidopa has virtually no effect on viability of melanoma cell cultures used by the authors. In the new version the authors confirmed it right in the Abstract and though carbidopa remains in the title of the article all the data presented correspond to the effects of rasagiline. At the same time the authors did not forget to discuss the phenomenon of the lack of effect of carbidopa on the viability of melanoma cells which they associated with the changes induced by carbidopa in tryptophan metabolism and the appearance of IAN.
Overally, it can be said that the article has been significantly revised and supplemented with several significant results based on BrdU-test and flow-cytometry to assess cell proliferation and analyse cell cycle stages being arrested. My comments were also taken into account. For instance, in the legend to Fig.3 the term selectivity index is explained more carefully “a quotient of IC50 values for malignant melanoma cell lines and IC50 for HaCaT,determined in the MTT assay” like I have reccomended. The lysing agent in the lysis buffer used to determine a 100% level of LDH in cells (Fig.4) is indicated as TX-100 in “LDH test” subsection of Materials and Methods. My opinion is that now the article appears to be far more interesting. A new mathematical approach, isobolographic analysis of drugs interaction makes it possible for the authors to distinguish synergistic, additive and antagonistic drug interactions and make important recommendations to cancer clinicians regarding the use of rasagiline together with MTX which exerted the most desirable synergistic effect on melanoma cell proliferation and should not use rasagiline in combination with CDDP. The article can be published in Cancer
A: Thank You for reviewing our manuscript. We are grateful for your valuable comments.
Round 2
Reviewer 1 Report (Previous Reviewer 4)
Comments and Suggestions for Authors
The revised version of paper titled (Analysis of antiviability effects of rasagiline and carbidopa on various human malignant melanoma cell lines: an in vitro isobolographic study.) By Authors Danuta Krasowska et al.
was improved compared to the originally submitted version. Iam glad to recommend accepting the current version R1 to publication in Cancers
Author Response
Thank you again for your time and valuable comments.
This manuscript is a resubmission of an earlier submission. The following is a list of the peer review reports and author responses from that submission.
Round 1
Reviewer 1 Report
Comments and Suggestions for Authors
Satisfactory
Author Response
C: Satisfactory
R: Thank you for your comment.
Reviewer 2 Report
Comments and Suggestions for Authors
In the present manuscript by Krasowska et al, the authors showed combinatorial effect of Rasagiline (used for PD treatment) and mitoxantrone (chemotherapeutic) in four malignant melanoma cell lines by performing cell viability test and Isobolographic analysis. Here are a few suggestions for the improvement of the manuscript-
1. The authors should mention the number of biological replicates done to generate each graph in all the figures in the main text.
2. The authors presented the IC50 values of Rasagiline in two different units (μg/mL and μM). I think the authors should maintain one unit.
3. The authors can include this reference (https://doi.org/10.3389/fphar.2019.01222) while describing isobolographic analysis.
4. The authors should include flow cytometry data discussing the anti-proliferative effect of Rasagiline alone (control), Rasagiline+CPPD and Rasagiline+MTX on the apoptosis induction (as discussed in author’s another paper Int. J. Mol. Sci. 2022, 23(14), 7653)
5. Do these cell lines differ in their expression level? Does this combinatorial effect of drugs depend on different expression level of these four cell lines used? If so, please add a discussion on this in the main text.
Author Response
In the present manuscript by Krasowska et al, the authors showed combinatorial effect of Rasagiline (used for PD treatment) and mitoxantrone (chemotherapeutic) in four malignant melanoma cell lines by performing cell viability test and Isobolographic analysis. Here are a few suggestions for the improvement of the manuscript-
R: Thank you for Your review. We hope that the changes made will be satisfactory.
- The authors should mention the number of biological replicates done to generate each graph in all the figures in the main text.
R: In the "Material and Methods" section, information was added about the fact that each experiment was performed in triplicate.
- The authors presented the IC50 values of Rasagiline in two different units (μg/mL and μM). I think the authors should maintain one unit.
R: As rasagiline stock and dosing were performed on μg/mL basis, in order to standardize the units for rasagiline, Table 1 presents the IC50 values in both, μg/mL and µM.
- The authors can include this reference (https://doi.org/10.3389/fphar.2019.01222) while describing isobolographic analysis.
R: The indicated publication has been added as suggested.
- The authors should include flow cytometry data discussing the anti-proliferative effect of Rasagiline alone (control), Rasagiline+CPPD and Rasagiline+MTX on the apoptosis induction (as discussed in author’s another paper Int. J. Mol. Sci. 2022, 23(14), 7653)
R: In the indicated publication, the effect of amantadine was assessed with flow cytometry. The results of the flow cytometry analysis also concerned only the compound administered by itself, but not in combinations.
- Do these cell lines differ in their expression level? Does this combinatorial effect of drugs depend on different expression level of these four cell lines used? If so, please add a discussion on this in the main text.
R: Two metastatic (FM55M2 and SK-MEL28) and two primary (FM55P and A375) cell cultures were used for the study. All tested cell lines in this study have the BRAF V600E mutation. Appropriate information is added in the text. The cell lines used herein did not differ in their expression level, therefore, there was no need to add such information to the discussion.
Reviewer 3 Report
Comments and Suggestions for Authors The article by Krasowska D. et al [v1] contains undoubtedly interesting information, but the title “Antiproliferative effects of rasagiline and carbidopa…” looks a little strangeconsidering the conclusion of the authors “Carbidopa at a concentration of 20 μM (maximal possible concentration in 0.1% DMSO) reduced melanoma cell proliferation by only
a few percent, - the main reason not to investigate its anti-proliferative effects in further experiments involving drug combinations.” So at least such small correction like “Analysis
of antiproliferative effects of …” makes the title more consistent with the results obtained. Moreover, may be better to transfer Fig.2 with no effects of carbidopa to Supplementary material. The meaning of the term selectivity index (Fig.3) should be explained more carefully, possibly like “a quotient of IC50 values for malignant melanoma cell lines and IC50 for HaCaT, determined in the MTT assay”. The lysing agent in the lysis buffer used to determine a 100% level of LDH in cells (Fig.4) should be indicated in “LDH test” subsection of Materials and Methods. Table 2 is strange because not in the text of the article, not in Materials and Methods there is any information about its content which is in fact outside the aims and content of the paper.
The authors at least must explain why they include such material in the paper. If the authors insist on including new information the title should be corrected, the results obtained must be reflected
in the Abstract and corresponding comments in Materials and Methods. As carbidopa did not demonstrate significant effects on melanoma cell lines used by the authors it looks more logic to transfer the discussion of its effects to the end of Discussion section
instead of its beginning. And also considering reference [39] the situation with low solubility of carbidopa looks strange. At the same time, the article itself is interesting. The authors use a new mathematical approach isobolographic analysis of drugs interaction which makes it possible to distinguish between
synergistic, additive and antagonistic drug interactions and to propose the most effective combinations in multitreatment scheme. The article can be published in Cancer after small revision.
[v1]
Author Response
The article by Krasowska D. et al [v1] contains undoubtedly interesting information, but the title “Antiproliferative effects of rasagiline and carbidopa…” looks a little strange
considering the conclusion of the authors “Carbidopa at a concentration of 20 μM (maximal possible concentration in 0.1% DMSO) reduced melanoma cell proliferation by only
a few percent, - the main reason not to investigate its anti-proliferative effects in further experiments involving drug combinations.” So at least such small correction like “Analysis
of antiproliferative effects of …” makes the title more consistent with the results obtained.
R: The title has been changed as suggested.
Moreover, may be better to transfer Fig.2 with no effects of carbidopa to Supplementary material. The meaning of the term selectivity index (Fig.3) should be explained more carefully, possibly like “a quotient of IC50 values for malignant melanoma cell lines and IC50 for HaCaT, determined in the MTT assay”.
R: The indicated information regarding the selectivity index has been added as requested.
The lysing agent in the lysis buffer used to determine a 100% level of LDH in cells (Fig.4) should be indicated in “LDH test” subsection of Materials and Methods.
R: Appropriate information has been added to the "LDH Test" section.
Table 2 is strange because not in the text of the article, not in Materials and Methods there is any information about its content which is in fact outside the aims and content of the paper. The authors at least must explain why they include such material in the paper. If the authors insist on including new information the title should be corrected, the results obtained must be reflected
in the Abstract and corresponding comments in Materials and Methods.
R: Tables 2 and 3 are the statistical part of the isobolographic analysis. Both tables are mentioned in the text of the isobolographic results section (on pages 8 and 9).
As carbidopa did not demonstrate significant effects on melanoma cell lines used by the authors it looks more logic to transfer the discussion of its effects to the end of Discussion section
instead of its beginning. And also considering reference [39] the situation with low solubility of carbidopa looks strange.
R: In our opinion, in the discussion we wanted to start with carbidopa as less important and then discuss rasagiline as more important in the publication. Regarding the citation [39] in the discussion the Authors explained that the used doses could be toxic to the cells due to the too high dose of DMSO.
At the same time, the article itself is interesting. The authors use a new mathematical approach isobolographic analysis of drugs interaction which makes it possible to distinguish between
synergistic, additive and antagonistic drug interactions and to propose the most effective combinations in multi treatment scheme. The article can be published in Cancer after small revision.
R: Thank you for Your review. We hope that the changes made will be satisfactory.
Reviewer 4 Report
Comments and Suggestions for Authors
Paper titled ( Antiproliferative effects of rasagiline and carbidopa (two anti- arkinson drugs) on various human malignant melanoma cell lines: an in vitro isobolographic study by Author Danuta Krasowska et al. ِactually I find this paper very weak and has no clinical potential. The rationale given by the authors to perform this study was not convincing and not correct. Kindly find below my detailed concerns:
1- Title: Antiproliferative effects of rasagiline and carbidopa (two anti- 2 Parkinson drugs) on various human malignant melanoma cell 3 lines: an in vitro isobolographic study
I find no need to mention (2 anti Parkinson drugs) and should be deleted.
Also carbidopa is not an antiparkinson drug !! it is a peripheral dopa decarboxylase inhibitor used in combination with levodopa
2- Abstract: authors wrote (RRasagiline and carbidopa as anti-Parkinson drugs alleviated symptoms of Par- 21 kinson’s disease and theoretically, they should possess anti-melanoma properties) I find this as an overestimated sentence. and also not convincing.
3- Abstract: should be amended by some numerical values from the results
4- The title gave me the impression that positive results were obtained for both rasagiline and carbidopa
Howver, upon reading the manuscript, I found this is not correct & hence the title must be revised to be informative
5- In introduction : line 74 (During clinical trials with rasagiline (a novel drug used to treat PD) : actually rasagiline is not a novel drug now
6- Many parts in the introduction do not have (enough) references although authors wrote (many studies)
7- Line 85: f carbidopa (a derivative of L-DOPA), this is not correct
8- Many missconceptions were found in this paper and in interpreting the results of previous studies as in introduction
9- Introduction: did not explore the rational or novelty of the study.
10- Methods section in general is too brief and lacks references MUST be extensively revised
11- Experiemntal design and animal grouping must be clearly explaine
12- Please give a title for study design & describe the groups in details in a clear way
13 -Ensure every abbreviation is explained at the first appearnace in abstract & then in the body text
14- Authors should give the source of chemicals, kits and antibodies completely and consistently (code, company, town, state and country) & version for software
15- Mention how drugs were prepared and solubilized
16- Authors have to check the normality of distribution of the results by a suitable post hoc test (such as Shapiro-Wilk test or K-S test) before deciding to choose certain ANOVA. If the normality test indicated normal dist of the data, so use one-way ANOVA, if not, use non parametric ANOVA. In all cases choose a suitable post-hoc test
17- Use appropriate abbreviations for minutes, seconds...etc
18- Every abbreviation in figures should be explained in the figure legend to be self explanatory & stands alone.
19- Authors should confirm in methods that "every possible comparison between the study groups was considered" and apply this in results
20- Mention "n" in each illustration individually
21- In methods: authors wrote (The IC50 values (for rasagiline + MTX and rasagiline + CDDP) 139 were statistically compared with the unpaired Student’s t test with Welch’s correction, as 140 suggested earlier [35].)
Please give reasons for selection of certain statistical procedures for each set of data
22- Give the limitations of this study
3on’s disease and theoretically, they should possess anti-melanoma properties
Author Response
Paper titled ( Antiproliferative effects of rasagiline and carbidopa (two anti- arkinson drugs) on various human malignant melanoma cell lines: an in vitro isobolographic study by Author Danuta Krasowska et al. ِactually I find this paper very weak and has no clinical potential. The rationale given by the authors to perform this study was not convincing and not correct. Kindly find below my detailed concerns:
R: Thank you for Your review. We hope that the changes made will be satisfactory.
1- Title: Antiproliferative effects of rasagiline and carbidopa (two anti- 2 Parkinson drugs) on various human malignant melanoma cell 3 lines: an in vitro isobolographic study
I find no need to mention (2 anti Parkinson drugs) and should be deleted.
Also carbidopa is not an antiparkinson drug !! it is a peripheral dopa decarboxylase inhibitor used in combination with levodopa
R: The title has been changed as suggested.
2- Abstract: authors wrote (Rasagiline and carbidopa as anti-Parkinson drugs alleviated symptoms of Parkinson’s disease and theoretically, they should possess anti-melanoma properties) I find this as an overestimated sentence. and also not convincing.
R: The mentioned sentence has been removed from the Abstract in order not to confuse the potential readers.
3- Abstract: should be amended by some numerical values from the results
R: All important numerical data are presented in the Abstract: i.e. IC50 for rasagiline and selectivity index.
4- The title gave me the impression that positive results were obtained for both rasagiline and carbidopa
Howver, upon reading the manuscript, I found this is not correct & hence the title must be revised to be informative
R: The title has been changed as suggested by the second and third Reviewers.
5- In introduction : line 74 (During clinical trials with rasagiline (a novel drug used to treat PD) : actually rasagiline is not a novel drug now
R: The sentence has been corrected.
6- Many parts in the introduction do not have (enough) references although authors wrote (many studies)
R: Citations have been appropriately supplemented in the text.
7- Line 85: f carbidopa (a derivative of L-DOPA), this is not correct
R: The sentence has been corrected.
8- Many missconceptions were found in this paper and in interpreting the results of previous studies as in introduction
R: The authors tried to correct all identified inaccuracies.
9- Introduction: did not explore the rational or novelty of the study.
R: The introduction clearly shows the connection between Parkinson's disease and melanoma. In addition, some research results are inconsistent, so it was decided to delve deeper into the topic.
10- Methods section in general is too brief and lacks references MUST be extensively revised
R: The section has been supplemented with many details, but some of them can still be found in the authors' previous publications.
11- Experimental design and animal grouping must be clearly explained
R: The publication described the results of in vitro studies, not animal studies. Presentation of animal grouping is irrelevant for this paper.
12- Please give a title for study design & describe the groups in details in a clear way
R: Information about the project number is provided. This is an internal project carried out at the Medical University of Lublin.
13 -Ensure every abbreviation is explained at the first appearnace in abstract & then in the body text
R: All abbreviations in the text of the manuscript have their own explanations.
14- Authors should give the source of chemicals, kits and antibodies completely and consistently (code, company, town, state and country) & version for software
R: Each reagent or cell line mentioned is accompanied by information regarding its origin.
15- Mention how drugs were prepared and solubilized
R: Appropriate information has been added to the text in the "Material and Methods" section
16- Authors have to check the normality of distribution of the results by a suitable post hoc test (such as Shapiro-Wilk test or K-S test) before deciding to choose certain ANOVA. If the normality test indicated normal dist of the data, so use one-way ANOVA, if not, use non parametric ANOVA. In all cases choose a suitable post-hoc test
R: Checking the normal distribution of the data is crucial for the entirely study and it is a standard in in vitro study. The normality was verified before the one-way ANOVA test was run. There is no need to explain such basic and fundamental information of statistical analysis of data. The Tukey’s post-hoc test mentioned in the Material and method section explains everything for readers keen on statistics.
17- Use appropriate abbreviations for minutes, seconds...etc
R: The text has been checked for corrections.
18- Every abbreviation in figures should be explained in the figure legend to be self explanatory & stands alone.
R: The figures have an appropriate legend.
19- Authors should confirm in methods that "every possible comparison between the study groups was considered" and apply this in results
R: “Every possible comparison between the study groups” or comparison of every group with every other groups makes no sense. The groups must be compared reasonably according to statistical standard ! Critical analysis of data which underwent statistical comparison in this study has been performed by a person fluent in statistics. The suggestion is irrelevant for this study.
20- Mention "n" in each illustration individually
R: In in vitro study there is no need to place “n” values in each illustration, especially if the experiments were performed in triplicate. Information on “n” values was presented in tables 2 and 3 (in parentheses). Presentation of “n” values on each column would make illustrations illegible. The suggestion is irrelevant for this study.
21- In methods: authors wrote (The IC50 values (for rasagiline + MTX and rasagiline + CDDP) 139 were statistically compared with the unpaired Student’s t test with Welch’s correction, as 140 suggested earlier [35].)
Please give reasons for selection of certain statistical procedures for each set of data
R: The Student’s t-test with Welch’s correction means that variances for the analyzed data, which underwent comparison has significantly differed one another. Such information is crucial for statistical analysis of data and it is clearly described in any Statistical textbook. There is no need to give reasons for applying such a statistical test. Checking the normal distribution of data and equal variances of the analyzed values is a standard procedure in statistics. There is no need to explain statistics in details in this study because the paper is about melanoma, but not about statistical analysis of data.
22- Give the limitations of this study
R: The final paragraph of the discussion concerns the limitations of the study.
Round 2
Reviewer 2 Report
Comments and Suggestions for Authors
The authors made required corrections. The paper can be accepted in present form.
Reviewer 4 Report
Comments and Suggestions for Authors
The revised version of paper titled (Antiproliferative effects of rasagiline and carbidopa (two anti-Parkinson drugs) on various human malignant melanoma cell lines: an in vitro isobolographic study) by Danuta Krasowska et al. submitted to cancers was very partly revised and still very weak and has no rationale. Authors did not offer enough point-to-point replies to the questions raised by the reviewer & did not mention where ( Page or line) we can follow this changes
Replies were very general and not specific & even did not improve the paper to reach the thrshold required for publication in Cancers
For example:
Question12- Please give a title for study design & describe the groups in details in a clear way
Authors replied:
R: Information about the project number is provided. This is an internal project carried out at the Medical University of Lublin.
Also author replied to this question:
21- In methods: authors wrote (The IC50 values (for rasagiline + MTX and rasagiline + CDDP) 139 were statistically compared with the unpaired Student’s t test with Welch’s correction, as 140 suggested earlier [35].)
Please give reasons for selection of certain statistical procedures for each set of data
R: The Student’s t-test with Welch’s correction means that variances for the analyzed data, which underwent comparison has significantly differed one another. Such information is crucial for statistical analysis of data and it is clearly described in any Statistical textbook. There is no need to give reasons for applying such a statistical test. Checking the normal distribution of data and equal variances of the analyzed values is a standard procedure in statistics. There is no need to explain statistics in details in this study because the paper is about melanoma, but not about statistical analysis of data.
I reply to the authors that statstical analysis is very crucial for study results and conclusion & your paper should be useful to the readers and educative so young researchers and other learn how to do a good methodlogy and statistical analysis
Another question
18- Every abbreviation in figures should be explained in the figure legend to be self explanatory & stands alone.
Authors replied
R: The figures have an appropriate legend.
I wish to inform authors that Reviewers provide their time and efforts to help researchers enhance their work to appear to the scientific community in a correct way and following the reviewers recommendations is usually very useful to researchers however, resistance to perform every recommendation is not the best situarion in my opinion
Also other replies to most of the questions cannot be followed in the manuscript
Thank you for your efforts